# Quantum Deformed Neural Networks

## Abstract

We develop a new quantum neural network layer designed to run efficiently on a quantum computer but that can be simulated on a classical computer when restricted in the way it entangles input states. We first ask how a classical neural network architecture, both fully connected or convolutional, can be executed on a quantum computer using quantum phase estimation. We then deform the classical layer into a quantum design which entangles activations and weights into quantum superpositions. While the full model would need the exponential speedups delivered by a quantum computer, a restricted class of designs represent interesting new classical network layers that still use quantum features. We show that these quantum deformed neural networks can be trained and executed on normal data such as images, and even classically deliver modest improvements over standard architectures.

## 1 Introduction

Quantum mechanics (QM) is the most accurate description for physical phenomena at very small scales, such as the behavior of molecules, atoms and subatomic particles. QM has a huge impact on our every day lives through technologies such as lasers, transistors (and thus microchips), superconductors and MRI.

A recent view of QM has formulated it as a (Bayesian) statistical methodology that only describes our subjective view of the (quantum) world, and how we update that view in light of evidence (i.e. measurements) (;t Hooft, 2016; Fuchs & Schack, 2013). This is in perfect analogy to the classical Bayesian view, a statistical paradigm extensively used in artificial intelligence where we maintain probabilities to represent our beliefs for events in the world.

The philosophy of this paper will be to turn this argument on its head. If we can view QM as just another consistent statistical theory that happens to describe nature at small scales, then we can also use this theory to describe *classical* signals by endowing them with a Hilbert space structure. In some sense, the 'only' difference with Bayesian statistics is that the positive probabilities are replaced with complex 'amplitudes'. This however has the dramatic effect that, unlike in classical statistics, interference between events now becomes a possibility. In this paper we show that this point of view uncovers new architectures and potential speedups for running neural networks on quantum computers.

We shall restrict our attention here to binary neural networks. We will introduce a new class of quantum neural networks and interpret them as generalizations of probabilistic binary neural networks, discussing potential speedups by running the models on a quantum computer. Then we will devise classically efficient algorithms to train the networks for a restricted set of quantum circuits. We present results of classical simulations of the quantum neural networks on real world data sizes and related gains in accuracy due to the quantum deformations. Contrary to almost all other works on quantum deep learning, our quantum neural networks can be simulated for practical classical problems, such as images or sound. The quantum nature of our models is there to increase the flexibility of the model-class and add new operators to the toolbox of the deep learning researcher, some of which may only reach their full potential when quantum computing becomes ubiquitous.

## 1.1 RELATED WORK

In Farhi & Neven (2018) variational quantum circuits that can be learnt via stochastic gradient descent were introduced. Their performance could be studied only on small input tasks such as classifying $4 \times 4$ images, due to the exponential memory requirement to simulate those circuits. Other works on variational quantum circuits for neural networks are Verdon et al. (2018); Beer et al. (2019). Their focus is similarly on the implementation on near term quantum devices and these models cannot be efficiently run on a classical computer. Exceptions are models which use tensor network simulations (Cong et al., 2019; Huggins et al., 2019) where the model can be scaled to $8 \times 8$ image data with 2 classes, at the price of constraining the geometry of the quantum circuit (Huggins et al., 2019). The quantum deformed neural networks introduced in this paper are instead a class of variational quantum circuits that can be scaled to the size of data that are used in traditional neural networks as we demonstrate in section 4.2.

Another line of work directly uses tensor networks as full precision machine learning models that can be scaled to the size of real data (Miles Stoudenmire & Schwab, 2016; Liu et al., 2017; Levine et al., 2017; Levine et al., 2019). However the constraints on the network geometry to allow for efficient contractions limit the expressivity and performance of the models. See however Cheng et al. (2020) for recent promising developments. Further, the tensor networks studied in these works are not unitary maps and do not directly relate to implementations on quantum computers.

A large body of work in quantum machine learning focuses on using quantum computing to provide speedups to classical machine learning tasks (Biamonte et al., 2017; Ciliberto et al., 2018; Wiebe et al., 2014), culminating in the discovery of quantum inspired speedups in classical algorithms (Tang, 2019). In particular, (Allcock et al., 2018; Cao et al., 2017; Schuld et al., 2015; Kerenidis et al., 2019) discuss quantum simulations of classical neural networks with the goal of improving the efficiency of classical models on a quantum computer. Our models differ from these works in two ways: i) we use quantum wave-functions to model weight uncertainty, in a way that is reminiscent of Bayesian models; ii) we design our network layers in a way that may only reach its full potential on a quantum computer due to exponential speedups, but at the same time can, for a restricted class of layer designs, be simulated on a classical computer and provide inspiration for new neural architectures. Finally, quantum methods for accelerating Bayesian inference have been discussed in Zhao et al. (2019b;a) but only for Gaussian processes while in this work we shall discuss relations to Bayesian neural networks.

## 2 GENERALIZED PROBABILISTIC BINARY NEURAL NETWORKS

Binary neural networks are neural networks where both weights and activations are binary. Let $\mathbb{B} = \{0, 1\}$. A fully connected binary neural network layer maps the $N_\ell$ activations $\boldsymbol{h}^{(\ell)}$ at level $\ell$ to the $N_{\ell+1}$ activations $\boldsymbol{h}^{(\ell+1)}$ at level $\ell + 1$ using weights $\boldsymbol{W}^{(\ell)} \in \mathbb{B}^{N_\ell N_{\ell+1}}$:

$$h_j^{(\ell+1)} = f(\boldsymbol{W}^{(\ell)}, \boldsymbol{h}^{(\ell)}) = \tau \left( \frac{1}{N_\ell + 1} \sum_{i=1}^{N_\ell} W_{j,i}^{(\ell)} h_i^{(\ell)} \right), \quad \tau(x) = \begin{cases} 0 & x < \frac{1}{2} \\ 1 & x \geq \frac{1}{2} \end{cases}. \quad (1)$$

We divide by $N_\ell + 1$ since the sum can take the $N_\ell + 1$ values $\{0, \dots, N_\ell\}$. We do not explicitly consider biases which can be introduced by fixing some activations to 1. In a classification model $\boldsymbol{h}^{(0)} = \boldsymbol{x}$ is the input and the last activation function is typically replaced by a softmax which produces output probabilities $p(\boldsymbol{y}|\boldsymbol{x}, \boldsymbol{W})$, where $\boldsymbol{W}$ denotes the collection of weights of the network.

Given $M$ input/output pairs $\boldsymbol{X} = (\boldsymbol{x}^1, \dots, \boldsymbol{x}^M), \boldsymbol{Y} = (\boldsymbol{y}^1, \dots, \boldsymbol{y}^M)$, a frequentist approach would determine the binary weights so that the likelihood $p(\boldsymbol{Y}|\boldsymbol{X}, \boldsymbol{W}) = \prod_{i=1}^{M} p(\boldsymbol{y}_i|\boldsymbol{x}_i, \boldsymbol{W})$ is maximized. Here we consider discrete or quantized weights and take the approach of variational optimization Staines & Barber (2012), which introduces a weight distribution $q_\theta(\boldsymbol{W})$ to devise a surrogate differential objective. For an objective $O(\boldsymbol{W})$, one has the bound $\max_{\boldsymbol{W} \in \mathbb{B}^N} O(\boldsymbol{W}) \geq \mathbb{E}_{q_\theta(\boldsymbol{W})}[O(\boldsymbol{W})]$, and the parameters of $q_\theta(\boldsymbol{W})$ are adjusted to maximize the lower bound. In our case we consider the objective:

$$\max_{\boldsymbol{W} \in \mathbb{B}^N} \log p(\boldsymbol{Y}|\boldsymbol{X}, \boldsymbol{W}) \geq \mathcal{L} := \mathbb{E}_{q_\theta(\boldsymbol{W})}[\log p(\boldsymbol{Y}|\boldsymbol{X}, \boldsymbol{W})] = \sum_{i=1}^{M} \mathbb{E}_{q_\theta(\boldsymbol{W})}[\log p(\boldsymbol{y}_i|\boldsymbol{x}_i, \boldsymbol{W})]. \quad (2)$$

While the optimal solution to equation 2 is a Dirac measure, one can add a regularization term $\mathcal{R}(\theta)$ to keep $q$ soft. In appendix A we review the connection with Bayesian deep learning, where $q_\theta(\boldsymbol{W})$ is the approximate posterior, $\mathcal{R}(\theta)$ is the KL divergence between $q_\theta(\boldsymbol{W})$ and the prior over weights, and the objective is derived by maximizing the evidence lower bound.

In both variational Bayes and variational optimization frameworks for binary networks, we have a variational distribution $q(\boldsymbol{W})$ and probabilistic layers where activations are random variables. We consider an approximate posterior factorized over the layers: $q(\boldsymbol{W}) = \prod_{\ell=1}^{L} q^{(\ell)}(\boldsymbol{W}^{(\ell)})$. If $\boldsymbol{h}^{(\ell)} \sim p^{(\ell)}$, equation 1 leads to the following recursive definition of distributions:

$$p^{(\ell+1)}(\boldsymbol{h}^{(\ell+1)}) = \sum_{\boldsymbol{h} \in \mathbb{B}^{N_\ell}} \sum_{\boldsymbol{W} \in \mathbb{B}^{N_\ell N_{\ell+1}}} \delta(\boldsymbol{h}^{(\ell+1)} - f(\boldsymbol{W}^{(\ell)}, \boldsymbol{h}^{(\ell)})) p^{(\ell)}(\boldsymbol{h}^{(\ell)}) q^{(\ell)}(\boldsymbol{W}^{(\ell)}). \quad (3)$$

We use the shorthand $p^{(\ell)}(\boldsymbol{h}^{(\ell)})$ for $p^{(\ell)}(\boldsymbol{h}^{(\ell)}|\boldsymbol{x})$ and the $\boldsymbol{x}$ dependence is understood. The average appearing in equation 2 can be written as an average over the network output distribution:

$$\mathbb{E}_{q_\theta(\boldsymbol{W})}[\log p(\boldsymbol{y}_i|\boldsymbol{x}_i, \boldsymbol{W})] = -\mathbb{E}_{p^{(L)}(\boldsymbol{h}^{(L)})}[g_i(\boldsymbol{y}_i, \boldsymbol{h}^{(L)})], \quad (4)$$

where the function $g_i$ is typically MSE for regression and cross-entropy for classification.

In previous works (Shayer et al., 2017; Peters & Welling, 2018), the approximate posterior was taken to be factorized: $q(\boldsymbol{W}^{(\ell)}) = \prod_{ij} q_{i,j}(W_{i,j}^{(\ell)})$, which results in a factorized activation distribution as well: $p^{(\ell)}(\boldsymbol{h}^{(\ell)}) = \prod_i p_i^{(\ell)}(h_i^{(\ell)})$. (Shayer et al., 2017; Peters & Welling, 2018) used the local reparameterization trick Kingma et al. (2015) to sample activations at each layer.

The quantum neural network we introduce below will naturally give a way to sample efficiently from complex distributions and in view of that we here generalize the setting: we act with a stochastic matrix $\boldsymbol{S}_\phi(\boldsymbol{h}', \boldsymbol{W}'|\boldsymbol{h}, \boldsymbol{W})$ which depends on parameters $\phi$ and correlates the weights and the input activations to a layer as follows:

$$\pi_{\phi,\theta}(\boldsymbol{h}', \boldsymbol{W}') = \sum_{\boldsymbol{h} \in \mathbb{B}^N} \sum_{\boldsymbol{W} \in \mathbb{B}^{NM}} \boldsymbol{S}_\phi(\boldsymbol{h}', \boldsymbol{W}'|\boldsymbol{h}, \boldsymbol{W}) p(\boldsymbol{h}) q_\theta(\boldsymbol{W}). \quad (5)$$

To avoid redundancy, we still take $q_\theta(\boldsymbol{W})$ to be factorized and let $\boldsymbol{S}$ create correlation among the weights as well. The choice of $\boldsymbol{S}$ will be related to the choice of a unitary matrix $\boldsymbol{D}$ in the quantum circuit of the quantum neural network. A layer is now made of the two operations, $\boldsymbol{S}_\phi$ and the layer map $f$, resulting in the following output distribution:

$$p^{(\ell+1)}(\boldsymbol{h}^{(\ell+1)}) = \sum_{\boldsymbol{h} \in \mathbb{B}_\ell^N} \sum_{\boldsymbol{W} \in \mathbb{B}^{N_\ell N_{\ell+1}}} \delta(\boldsymbol{h}^{(\ell+1)} - f(\boldsymbol{W}^{(\ell)}, \boldsymbol{h}^{(\ell)})) \pi_{\phi,\theta}^{(\ell)}(\boldsymbol{h}^{(\ell)}, \boldsymbol{W}^{(\ell)}), \quad (6)$$

which allows one to compute the network output recursively. Both the parameters $\phi$ and $\theta$ will be learned to solve the following optimization problem:

$$\min_{\theta, \phi} \mathcal{R}(\theta) + \mathcal{R}'(\phi) - \mathcal{L}. \quad (7)$$

where $\mathcal{R}(\theta), \mathcal{R}'(\phi)$ are regularization terms for the parameters $\theta, \phi$. We call this model a generalized probabilistic binary neural network, with $\phi$ deformation parameters chosen such that $\phi = 0$ gives back the standard probabilistic binary neural network.

To study this model on a classical computer we need to choose $S$ which leads to an efficient sampling algorithm for $\pi_{\phi,\theta}$. In general, one could use Markov Chain Monte Carlo, but there exists situations for which the mixing time of the chain grows exponentially in the size of the problem (Levin & Peres, 2017). In the next section we will show how quantum mechanics can enlarge the set of probabilistic binary neural networks that can be efficiently executed and in the subsequent sections we will show experimental results for a restricted class of correlated distributions inspired by quantum circuits that can be simulated classically.

## 3 QUANTUM IMPLEMENTATION

Quantum computers can sample from certain correlated distributions more efficiently than classical computers (Aaronson & Chen, 2016; Arute et al., 2019). In this section, we devise a quantum circuit

that implements the generalized probabilistic binary neural networks introduced above, encoding $\pi_{\theta,\phi}$ in a quantum circuit. This leads to an exponential speedup for running this model on a quantum computer, opening up the study of more complex probabilistic neural networks.

A quantum implementation of a binary perceptron was introduced in Schuld et al. (2015) as an application of the quantum phase estimation algorithm (Nielsen & Chuang, 2000). However, no quantum advantage of the quantum simulation was shown. Here we will extend that result in several ways: i) we will modify the algorithm to represent the generalized probabilistic layer introduced above, showing the quantum advantage present in our setting; ii) we will consider the case of multi layer percetrons as well as convolutional networks.

## 3.1 INTRODUCTION TO QUANTUM NOTATION AND QUANTUM PHASE ESTIMATION

As a preliminary step, we introduce notations for quantum mechanics. We refer the reader to Appendix B for a more thorough review of quantum mechanics. A qubit is the vector space of normalized vectors $|\psi\rangle \in \mathbb{C}^2$. $N$ qubits form the set of unit vectors in $(\mathbb{C}^2)^{\otimes N} \cong \mathbb{C}^{2^N}$ spanned by all $N$-bit strings, $|b_1, \ldots, b_N\rangle \equiv |b_1\rangle \otimes \cdots \otimes |b_N\rangle$, $b_i \in \mathbb{B}$. Quantum circuits are unitary matrices on this space. The probability of a measurement with outcome $\phi_i$ is given by matrix element of the projector $|\phi_i\rangle \langle \phi_i|$ in a state $|\psi\rangle$, namely $p_i = \langle \psi | \phi_i \rangle \langle \phi_i | \psi \rangle = |\langle \phi_i | \psi \rangle|^2$, a formula known as Born's rule.

Next, we describe the quantum phase estimation (QPE), a quantum algorithm to estimate the eigenphases of a unitary $\boldsymbol{U}$. Denote the eigenvalues and eigenvectors of $\boldsymbol{U}$ by $\exp\left(\frac{2\pi i}{2^t}\varphi_\alpha\right)$ and $|v_\alpha\rangle$, and assume that the $\varphi_\alpha$'s can be represented with a finite number $t$ of bits: $\varphi_\alpha = 2^{t-1}\varphi_\alpha^1 + \cdots + 2^0\varphi_\alpha^t$. (This is the case of relevance for a binary network.) Then introduce $t$ ancilla qubits in state $|0\rangle^{\otimes t}$. Given an input state $|\psi\rangle$, QPE is the following unitary operation:

$$|0\rangle^{\otimes t} \otimes |\psi\rangle \xrightarrow{\text{QPE}} \sum_\alpha \langle v_\alpha | \psi \rangle \, |\varphi_\alpha\rangle \otimes |v_\alpha\rangle \ . \tag{8}$$

Appendix B.1 reviews the details of the quantum circuit implementing this map, whose complexity is linear in $t$. Now using the notation $\tau$ for the threshold non-linearity introduced in equation 1, and recalling the expansion $2^{-t}\varphi = 2^{-1}\varphi^1 + \cdots + 2^{-t}\varphi^t$, we note that if the first bit $\varphi^1 = 0$ then $2^{-t}\varphi < \frac{1}{2}$ and $\tau(2^{-t}\varphi) = 0$, while if $\varphi^1 = 1$, then $2^{-t}\varphi \geq \frac{1}{2}$ and $\tau(2^{-t}\varphi) = 1$. In other words, $\delta_{\varphi^1,b} = \delta_{\tau(2^{-t}\varphi),b}$ and the probability $p(b)$ that after the QPE the first ancilla bit is $b$ is given by:

$$\left(\sum_\alpha \overline{\langle v_\alpha | \psi \rangle} \, \langle \varphi_\alpha | \otimes \langle v_\alpha |\right)\left[|b\rangle\langle b| \otimes \mathbf{1}\right]\left(\sum_\beta \langle v_\beta | \psi \rangle \, |\varphi_\beta\rangle \otimes |v_\beta\rangle\right) = \sum_\alpha |\langle v_\alpha | \psi \rangle|^2 \delta_{\tau(2^{-t}\varphi_\alpha),b} \, , \quad (9)$$

where $\left[|b\rangle\langle b| \otimes \mathbf{1}\right]$ is an operator that projects the first bit to the state $|b\rangle$ and leaves the other bits untouched.

## 3.2 DEFINITION AND ADVANTAGES OF QUANTUM DEFORMED NEURAL NETWORKS

Armed with this background, we can now apply quantum phase estimation to compute the output of the probabilistic layer of equation 6. Let $N$ be the number of input neurons and $M$ that of output neurons. We introduce qubits to represent inputs and weights bits:

$$|\boldsymbol{h}, \boldsymbol{W}\rangle \in \mathcal{V}_h \otimes \mathcal{V}_W \, , \quad \mathcal{V}_h = \bigotimes_{i=1}^N (\mathbb{C}^2)_i \, , \quad \mathcal{V}_W = \bigotimes_{i=1}^N \bigotimes_{j=1}^M (\mathbb{C}^2)_{ij} \, . \tag{10}$$

Then we introduce a Hamiltonian $\boldsymbol{H}_j$ acting non trivially only on the $N$ input activations and the $N$ weights at the $j$-th row:

$$\boldsymbol{H}_j = \sum_{i=1}^N \boldsymbol{B}_{ji}^W \boldsymbol{B}_i^h \, , \tag{11}$$

and $\boldsymbol{B}_i^h$ ($\boldsymbol{B}_{ji}^W$) is the matrix $\boldsymbol{B} = |1\rangle \langle 1|$ acting on the $i$-th activation ($ji$-th weight) qubit. Note that $\boldsymbol{H}_j$ singles out terms from the state $|\boldsymbol{h}, \boldsymbol{W}\rangle$ where both $h_j = 1$ and $W_{ij} = 1$ and then adds them

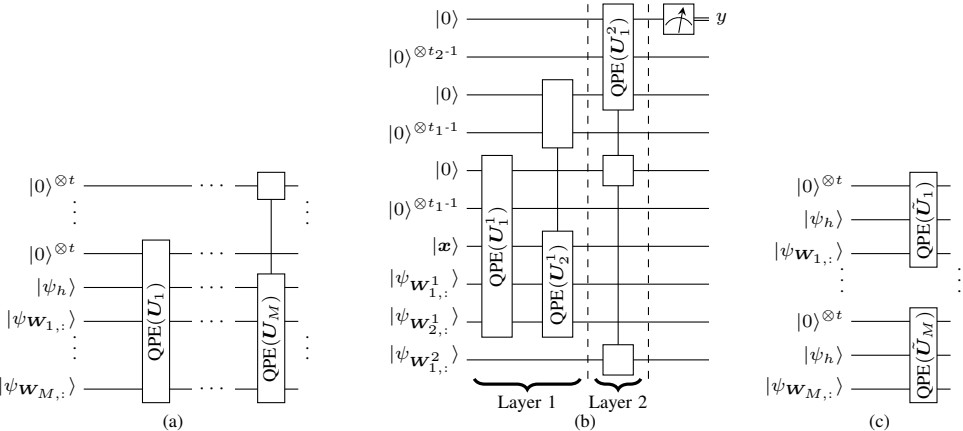

Figure 1: (a) Quantum circuit implementing a quantum deformed layer. The thin vertical line indicates that the gate acts as identity on the wires crossed by the line. (b) Quantum deformed multilayer perceptron with 2 hidden quantum neurons and 1 output quantum neuron. $|\boldsymbol{x}\rangle$ is an encoding of the input signal, $y$ is the prediction. The superscript $\ell$ in $\boldsymbol{U}_j^\ell$ and $\boldsymbol{W}_{j,:}^\ell$ refers to layer $\ell$. We split the blocks of $t_\ell$ ancilla qubits into a readout qubit that encodes the layer output amplitude and the rest. (c) Modification of a layer for classical simulations.

up, i.e. the eigenvalues of $\boldsymbol{H}_j$ are the preactivations of equation 1:

$$\boldsymbol{H}_j \left|\boldsymbol{h}, \boldsymbol{W}\right\rangle = \varphi(\boldsymbol{h}, \boldsymbol{W}_{j,:}) \left|\boldsymbol{h}, \boldsymbol{W}\right\rangle , \quad \varphi(\boldsymbol{h}, \boldsymbol{W}_{j,:}) = \sum_{i=1}^{N} W_{ji} h_i . \tag{12}$$

Now define the unitary operators:

$$\boldsymbol{U}_j = \boldsymbol{D} \mathrm{e}^{\frac{2\pi i}{N+1} \boldsymbol{H}_j} \boldsymbol{D}^{-1} , \tag{13}$$

where $\boldsymbol{D}$ is another generic unitary, and as we shall see shortly, its eigenvectors will be related to the entries of the classical stochastic matrix $\boldsymbol{S}$ in section 2. Since $\boldsymbol{U}_j \boldsymbol{U}_{j'} = \boldsymbol{D} \mathrm{e}^{\frac{2\pi i}{N+1}(\boldsymbol{H}_j + \boldsymbol{H}_{j'})} \boldsymbol{D}^{-1} = \boldsymbol{U}_{j'} \boldsymbol{U}_j$, we can diagonalize all the $\boldsymbol{U}_j$'s simultaneously and since they are conjugate to $\mathrm{e}^{\frac{2\pi i}{N+1} \boldsymbol{H}_j}$ they will have the same eigenvalues. Introducing the eigenbasis $\left|\boldsymbol{h}, \boldsymbol{W}\right\rangle_{\boldsymbol{D}} = \boldsymbol{D} \left|\boldsymbol{h}, \boldsymbol{W}\right\rangle$, we have:

$$\boldsymbol{U}_j \left|\boldsymbol{h}, \boldsymbol{W}\right\rangle_{\boldsymbol{D}} = \mathrm{e}^{\frac{2\pi i}{N+1} \varphi(\boldsymbol{h}, \boldsymbol{W}_{j,:})} \left|\boldsymbol{h}, \boldsymbol{W}\right\rangle_{\boldsymbol{D}} . \tag{14}$$

Note that $\varphi \in \{0, \ldots, N\}$ so we can represent it with exactly $t$ bits, $N = 2^t - 1$. Then we add $M$ ancilla resources, each of $t$ qubits, and sequentially perform $M$ quantum phase estimations, one for each $\boldsymbol{U}_j$, as depicted in figure 1 (a). We choose the following input state

$$\left|\psi\right\rangle = \left|\psi\right\rangle_h \otimes \bigotimes_{j=1}^{M} \left|\psi\right\rangle_{\boldsymbol{W}_{j,:}} , \quad \left|\psi\right\rangle_{\boldsymbol{W}_{j,:}} = \bigotimes_{i=1}^{N} \left[ \sqrt{q_{ji}(W_{ji} = 0)} \left|0\right\rangle + \sqrt{q_{ji}(W_{ji} = 1)} \left|1\right\rangle \right] , \tag{15}$$

where we have chosen the weight input state according to the factorized variational distribution $q_{ij}$ introduced in section 2. In fact, this state corresponds to the following probability distribution via Born's rule:

$$p(\boldsymbol{h}, \boldsymbol{W}) = |\left\langle \boldsymbol{h}, \boldsymbol{W} | \psi \right\rangle|^2 = p(\boldsymbol{h}) \prod_{j=1}^{M} \prod_{i=1}^{N} q_{ji}(W_{ji}), \quad p(\boldsymbol{h}) = |\left\langle \boldsymbol{h} | \psi \right\rangle_h|^2 . \tag{16}$$

The state $\left|\psi_h\right\rangle$ is discussed below. Now we show that a non-trivial choice of $\boldsymbol{D}$ leads to an effective correlated distribution. The $j$-th QPE in figure 1 (a) corresponds to equation 8 where we identify $\left|v_\alpha\right\rangle \equiv \left|\boldsymbol{h}, \boldsymbol{W}\right\rangle_{\boldsymbol{D}}, \left|\varphi_\alpha\right\rangle \equiv \left|\varphi(\boldsymbol{h}, \boldsymbol{W}_{j,:})\right\rangle$ and we make use of the $j$-th block of $t$ ancillas. After $M$ steps we compute the outcome probability of a measurement of the first qubit in each of the $M$ registers of the ancillas. We can extend equation 9 to the situation of measuring multiple qubits, and recalling that the first bit of an integer is the most significant bit, determining whether $2^{-t} \varphi(\boldsymbol{h}, \boldsymbol{W}_{j,:}) = (N+1)^{-1} \varphi(\boldsymbol{h}, \boldsymbol{W}_{j,:})$ is greater or smaller than $1/2$, the probability of outcome $\boldsymbol{h}' = (h_1', \ldots, h_M')$ is

$$p(\boldsymbol{h}') = \sum_{\boldsymbol{h} \in \mathbb{B}^N} \sum_{\boldsymbol{W} \in \mathbb{B}^{NM}} \delta_{\boldsymbol{h}', f(\boldsymbol{W}, \boldsymbol{h})} |\left\langle \psi | \boldsymbol{h}, \boldsymbol{W}\right\rangle_{\boldsymbol{D}}|^2 , \tag{17}$$

where $f$ is the layer function introduced in equation 1. We refer to appendix C for a detailed derivation. Equation 17 is the generalized probabilistic binary layer introduced in equation 6 where $\boldsymbol{D}$ corresponds to a non-trivial $\boldsymbol{S}$ and a correlated distribution when $\boldsymbol{D}$ entangles the qubits:

$$\pi(\boldsymbol{h}, \boldsymbol{W}) = |\langle\psi| \boldsymbol{D} |\boldsymbol{h}, \boldsymbol{W}\rangle|^2 . \tag{18}$$

The variational parameters $\phi$ of $\boldsymbol{S}$ are now parameters of the quantum circuit $\boldsymbol{D}$. Sampling from $\pi$ can be done by doing repeated measurements of the first $M$ ancilla qubits of this quantum circuit. On quantum hardware $\mathrm{e}^{\frac{2\pi i}{N+1}\boldsymbol{H}_j}$ can be efficiently implemented since it is a product of diagonal two-qubits quantum gates. We shall consider unitaries $\boldsymbol{D}$ which have efficient quantum circuit approximations. Then computing the quantum deformed layer output on a quantum computer is going to take time $O(tMu(N))$ where $u(N)$ is the time it takes to compute the action of $\boldsymbol{U}_j$ on an input state. There exists $\boldsymbol{D}$ such that sampling from equation 18 is exponentially harder classically than quantum mechanically, a statement forming the basis for quantum supremacy experiments on noisy, intermediate scale quantum computers (Aaronson & Chen, 2016; Arute et al., 2019). Examples are random circuits with two-dimensional entanglement patterns, which from a machine learning point of view can be natural when considering image data. Other examples are $\boldsymbol{D}$ implementing time evolution operators of physical systems, whose simulation is exponentially hard classically, resulting in hardness of sampling from the time evolved wave function. Quantum supremacy experiments give foundations to which architectures can benefit from quantum speedups, but we remark that the proposed quantum architecture, which relies on quantum phase estimation, is designed for error-corrected quantum computers.

Even better, on quantum hardware we can avoid sampling intermediate activations altogether. At the first layer, the input can be prepared by encoding the input bits in the state $|\boldsymbol{x}\rangle$. For the next layers, we simply use the output state as the input to the next layer. One obtains thus the quantum network of figure 1 (b) and the algorithm for a layer is summarized in procedure 1. Note that all the qubits associated to the intermediate activations are entangled. Therefore the input state $|\psi_h\rangle$ would have to be replaced by a state in $\mathcal{V}_h$ plus all the other qubits, where the gates at the next layer would act only on $\mathcal{V}_h$ in the manner described in this section. (An equivalent and more economical mathematical description is to use the reduced density matrix $\rho_h$ as input state.) We envision two other possible procedures for what happens after the first layer: i) we sample from equation 17 and initialize $|\psi_h\rangle$ to the bit string sampled in analogy to the classical quantization of activations; ii) we sample many times to reconstruct the classical distribution and encode it in $|\psi_h\rangle$. In our classical simulations below we will be able to actually calculate the probabilities and can avoid sampling.

Finally, we remark that at present it is not clear whether the computational speedup exhibited by our architecture translates to a learning advantage. This is an outstanding question whose full answer will require an empirical evaluation with a quantum computer. Next, we will try to get as close as possible to answer this question by studying a quantum model that we can simulate classically.

### 3.3 MODIFICATIONS FOR CLASSICAL SIMULATIONS

In this paper we will provide classical simulations of the quantum neural networks introduced above for a restricted class of designs. We do this for two reasons: first to convince the reader that the quantum layers hold promise (even though we can not simulate the proposed architecture in its full glory due to the lack of access to a quantum computer) and second, to show that these ideas can be interesting as new designs, even "classically" (by which we mean architectures that can be executed on a classical computer).

To parallelize the computations for different output neurons, we do the modifications to the setup just explained which are depicted in figure 1 (c). We clone the input activation register $M$ times, an operation that quantum mechanically is only approximate (Nielsen & Chuang, 2000) but exact classically. Then we associate the $j$-th copy to the $j$-th row of the weight matrix, thus forming pairs for each $j = 1, \ldots, M$:

$$|\boldsymbol{h}, \boldsymbol{W}_{j,:}\rangle \in \mathcal{V}_h \otimes \mathcal{V}_{W,j} , \quad \mathcal{V}_{W,j} = \bigotimes_{i=1}^{N}(\mathbb{C}^2)_{ji} \tag{19}$$

Fixing $j$, we introduce the unitary $\mathrm{e}^{\frac{2\pi i}{N+1}\boldsymbol{H}_j}$ diagonal in the basis $|\boldsymbol{h}, \boldsymbol{W}_{j,:}\rangle$ as in equation 11 and define the new unitary:

$$\tilde{\boldsymbol{U}}_j = \boldsymbol{D}_j \mathrm{e}^{\frac{2\pi i}{N+1}\boldsymbol{H}_j} \boldsymbol{D}_j^{-1} \,, \qquad (20)$$

where w.r.t. equation 13 we now let $\boldsymbol{D}_j$ depend on $j$. We denote the eigenvectors of $\tilde{\boldsymbol{U}}_j$ by $|\boldsymbol{h}, \boldsymbol{W}_{j,:}\rangle_{\boldsymbol{D}_j} = \boldsymbol{D}_j |\boldsymbol{h}, \boldsymbol{W}_{j,:}\rangle$ and the eigenvalue is $\varphi(\boldsymbol{h}, \boldsymbol{W}_{j,:})$ introduced in equation 12. Supposing that we know $p(\boldsymbol{h}) = \prod_i p_i(h_i)$, we apply the quantum phase estimation to $\tilde{\boldsymbol{U}}_j$ with input:

$$|\psi_j\rangle = |\psi_h\rangle \otimes |\psi\rangle_{\boldsymbol{W}_{j,:}} \,, \quad |\psi_h\rangle = \bigotimes_{i=1}^{N} \left[ \sqrt{p_i(h_i = 0)} \, |0\rangle + \sqrt{p_i(h_i = 1)} \, |1\rangle \right] \,, \qquad (21)$$

and $|\psi\rangle_{\boldsymbol{W}_{j,:}}$ is defined in equation 15. Going through similar calculations as those done above shows that measurements of the first qubit will be governed by the probability distribution of equation 6 factorized over output channels since the procedure does not couple them: $\pi(\boldsymbol{h}, \boldsymbol{W}) = \prod_{j=1}^{M} |\langle\psi_j| \boldsymbol{D}_j |\boldsymbol{h}, \boldsymbol{W}_{j,:}\rangle|^2$. So far, we have focused on fully connected layers. We can extend the derivation of this section to the convolution case, by applying the quantum phase estimation on images patches of the size equal to the kernel size as explained in appendix D.

## 4 CLASSICAL SIMULATIONS FOR LOW ENTANGLEMENT

### 4.1 THEORY

It has been remarked in (Shayer et al., 2017; Peters & Welling, 2018) that when the weight and activation distributions at a given layer are factorized, $p(\boldsymbol{h}) = \prod_i p_i(h_i)$ and $q(\boldsymbol{W}) = \prod_{ij} q_{ij}(W_{ij})$, the output distribution in equation 3 can be efficiently approximated using the central limit theorem (CLT). The argument goes as follows: for each $j$ the preactivations $\varphi(\boldsymbol{h}, \boldsymbol{W}_{j,:}) = \sum_{i=1}^{N} W_{j,i} h_i$ are sums of independent binary random variables $W_{j,i} h_i$ with mean and variance:

$$\mu_{ji} = \mathbb{E}_{w\sim q_{ji}}(w)\mathbb{E}_{h\sim p_i}(h) \,, \quad \sigma_{ji}^2 = \mathbb{E}_{w\sim q_{ji}}(w^2)\mathbb{E}_{h\sim p_i}(h^2) - \mu_{ji}^2 = \mu_{ji}(1 - \mu_{ji}) \,, \qquad (22)$$

We used $b^2 = b$ for a variable $b \in \{0, 1\}$. The CLT implies that for large $N$ we can approximate $\varphi(\boldsymbol{h}, \boldsymbol{W}_{j,:})$ with a normal distribution with mean $\mu_j = \sum_i \mu_{ji}$ and variance $\sigma_j^2 = \sum_i \sigma_{ji}^2$. The distribution of the activation after the non-linearity of equation 1 can thus be computed as:

$$p(\tau(\tfrac{1}{N+1}\varphi(\boldsymbol{h}, \boldsymbol{W}_{j,:})) = 1) = p(2\varphi(\boldsymbol{h}, \boldsymbol{W}_{j,:}) - N > 0) = \Phi\left(-\frac{2\mu_j - N}{2\sigma_j}\right) \,, \qquad (23)$$

$\Phi$ being the CDF of the standard normal distribution. Below we fix $j$ and omit it for notation clarity.

As reviewed in appendix B, commuting observables in quantum mechanics behave like classical random variables. The observable of interest for us, $\boldsymbol{D}\boldsymbol{H}\boldsymbol{D}^{-1}$ of equation 20, is a sum of commuting terms $\boldsymbol{K}_i \equiv \boldsymbol{D}\boldsymbol{B}_i^W \boldsymbol{B}_i^h \boldsymbol{D}^{-1}$ and if their joint probability distribution is such that these random variables are weakly correlated, i.e.

$$\langle\psi| \boldsymbol{K}_i \boldsymbol{K}_{i'} |\psi\rangle - \langle\psi| \boldsymbol{K}_i |\psi\rangle \langle\psi| \boldsymbol{K}_{i'} |\psi\rangle \to 0 \,, \quad \text{if } |i - i'| \to \infty \,, \qquad (24)$$

then the CLT for weakly correlated random variables applies, stating that measurements of $\boldsymbol{D}\boldsymbol{H}\boldsymbol{D}^{-1}$ in state $|\psi\rangle$ are governed by a Gaussian distribution $\mathcal{N}(\mu, \sigma^2)$ with

$$\mu = \langle\psi| \boldsymbol{D}\boldsymbol{H}\boldsymbol{D}^{-1} |\psi\rangle \,, \quad \sigma^2 = \langle\psi| \boldsymbol{D}\boldsymbol{H}^2\boldsymbol{D}^{-1} |\psi\rangle - \mu^2 \,. \qquad (25)$$

Finally, we can plug these values into equation 23 to get the layer output probability.

We have cast the problem of simulating the quantum neural network to the problem of computing the expectation values in equation 25. In physical terms, these are related to correlation functions of $\boldsymbol{H}$ and $\boldsymbol{H}^2$ after evolving a state $|\psi\rangle$ with the operator $\boldsymbol{D}$. These can be efficiently computed classically for one dimensional and lowly entangled quantum circuits $\boldsymbol{D}$ (Vidal, 2003). In view of that here we consider a 1d arrangement of activation and weight qubits, labeled by $i = 0, \dots, 2N - 1$, where the even qubits are associated with activations and the odd are associated with weights. We then choose:

$$\boldsymbol{D} = \prod_{i=0}^{N-1} \boldsymbol{Q}_{2i,2i+1} \prod_{i=0}^{N-1} \boldsymbol{P}_{2i+1,2i+2} \,, \qquad (26)$$

where $\boldsymbol{Q}_{2i,2i+1}$ acts non-trivially on qubits $2i, 2i+1$, i.e. onto the $i$-th activation and $i$-th weight qubits, while $\boldsymbol{P}_{2i,2i+1}$ on the $i$-th weight and $i+1$-th activation qubits. We depict this quantum circuit in figure 2 (a). As explained in detail in appendix E, the computation of $\mu$ involves the matrix element of $\boldsymbol{K}_i$ in the product state $|\psi\rangle$ while $\sigma^2$ involves that of $\boldsymbol{K}_i\boldsymbol{K}_{i+1}$. Due to the structure of $\boldsymbol{D}$, these operators act locally on 4 and 6 sites respectively as depicted in figure 2 (b)-(c). This

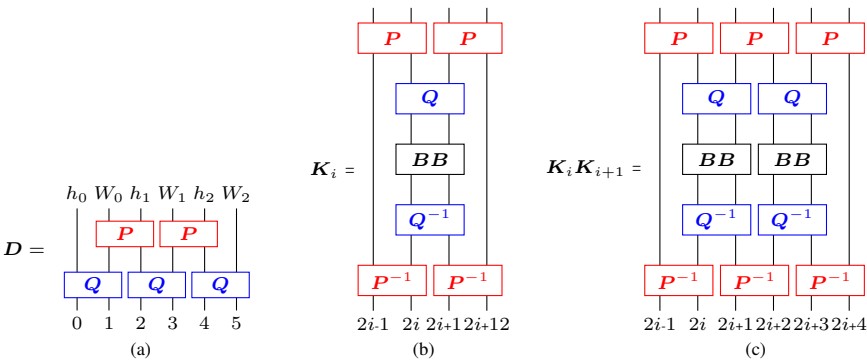

Figure 2: (a) The entangling circuit $\boldsymbol{D}$ for $N = 3$. (b) $\boldsymbol{K}_i$ entering the computation of $\mu$. (c) $\boldsymbol{K}_i\boldsymbol{K}_{i+1}$ entering the computation of $\sigma^2$. Indices on $\boldsymbol{P}, \boldsymbol{Q}, \boldsymbol{B}$ are omitted for clarity. Time flows downwards.

implies that the computation of equation 25, and so of the full layer, can be done in $O(N)$ and easily parallelized. Appendix E contains more details on the complexity, while Procedure 2 describes the algorithm for the classical simulation discussed here.

---

**Procedure 1** Quantum deformed layer. $\mathrm{QPE}_j(\boldsymbol{U}, \mathcal{I})$ is quantum phase estimation for a unitary $\boldsymbol{U}$ acting on the set $\mathcal{I}$ of activation qubits and the $j$-th weights/ancilla qubits. $\boldsymbol{H}_j$ is in equation 11.

---

**Input:** $\{q_{ij}\}_{i=0,\dots,N-1}^{j=0,\dots,M-1}, |\psi\rangle, \mathcal{I}, \boldsymbol{D}, t$
**Output:** $|\psi\rangle$
  **for** $j = 0$ **to** $M - 1$ **do**
    $|\psi\rangle_{\boldsymbol{W}_{j,:}} \leftarrow \bigotimes_{i=1}^{N} \left[ \sqrt{q_{ji}} |0\rangle + \sqrt{1 - q_{ji}} |1\rangle \right]$
    $|\psi\rangle \leftarrow |0\rangle^{\otimes t} \otimes |\psi\rangle \otimes |\psi\rangle_{\boldsymbol{W}_{j,:}}$
    $\boldsymbol{U} \leftarrow \boldsymbol{D}\mathrm{e}^{\frac{2\pi i}{N+1}\boldsymbol{H}_j}\boldsymbol{D}^{-1}$ {This requires to approximate the unitary with quantum gates}
    $|\psi\rangle \leftarrow \mathrm{QPE}_j(\boldsymbol{U}, \mathcal{I}) |\psi\rangle$
  **end for**

---

**Procedure 2** Classical simulation of a quantum deformed layer with $N$ ($M$) inputs (outputs).

---

**Input:** $\{q_{ij}\}_{i=0,\dots,N-1}^{j=0,\dots,M-1}, \{p_i\}_{i=0,\dots,N-1}, \boldsymbol{P} = \{\boldsymbol{P}_{2i-1,2i}^j\}_{i=1,\dots,N-1}^{j=0,\dots,M-1}, \boldsymbol{Q} = \{\boldsymbol{Q}_{2i,2i+1}^j\}_{i=0,\dots,N-1}^{j=0,\dots,M-1}$
**Output:** $\{p_i'\}_{i=1,\dots,M}$
  **for** $j = 0$ **to** $M - 1$ **do**
    **for** $i = 0$ **to** $N - 1$ **do**
      $\psi_{2i} \leftarrow \left[ \sqrt{p_i}, \sqrt{1 - p_i} \right]$
      $\psi_{2i+1} \leftarrow \left[ \sqrt{q_{ij}}, \sqrt{1 - q_{ij}} \right]$
    **end for**
    **for** $i = 0$ **to** $N - 1$ **do**
      $\mu_i \leftarrow \mathrm{computeMu}(i, \psi, \boldsymbol{P}, \boldsymbol{Q})$ {This implements equation 45 of appendix E}
      $\gamma_{i,i+1} \leftarrow \mathrm{computeGamma}(i, \psi, \boldsymbol{P}, \boldsymbol{Q})$ {This implements equation 49 of appendix E}
    **end for**
    $\mu \leftarrow \sum_{i=0}^{N-1} \mu_i$
    $\sigma^2 \leftarrow 2\sum_{i=0}^{N-2}(\gamma_{i,i+1} - \mu_i\mu_{i+1}) + \sum_{i=0}^{N-1}(\mu_i - \mu_i^2)$
    $p_j' \leftarrow \Phi\left(-\frac{2\mu - N}{2\sqrt{\sigma^2}}\right)$
  **end for**

---

## 4.2 EXPERIMENTS

We present experiments for the model of the previous section. At each layer, $q_{ij}$ and $\boldsymbol{D}_j$ are learnable. They are optimized to minimize the loss of equation 7 where following (Peters & Welling, 2018; Shayer et al., 2017) we take $\mathcal{R} = \beta \sum_{\ell,i,j} q_{ij}^{(\ell)}(1 - q_{ij}^{(\ell)})$, and $\mathcal{R}'$ is the $L_2$ regularization loss of the parameters of $\boldsymbol{D}_j$. $\mathcal{L}$ coincides with equation 2. We implemented and trained several architectures with different deformations. Table 1 contains results for two standard image datasets, MNIST and Fashion MNIST. Details of the experiments are in appendix F. The classical baseline is based on (Peters & Welling, 2018), but we use fewer layers to make the simulation of the deformation cheaper and use no batch norm, and no max pooling.

The general deformation ([PQ]) performs best in all cases. In the simplest case of a single dense layer (A), the gain is $+3.2\%$ for MNIST and $+2.6\%$ for Fashion MNIST on test accuracy. For convnets, we could only simulate a single deformed layer due to computational issues and the gain is around or less than $1\%$. We expect that deforming all layers will give a greater boost as the improvements diminish with decreasing the ratio of deformation parameters over classical parameters ($q_{ij}$). The increase in accuracy comes at the expense of more parameters. In appendix F we present additional results showing that quantum models can still deliver modest accuracy improvement w.r.t. convolutional networks with the same number of parameters.

Table 1: Test accuracies for MNIST and Fashion MNIST. With the notation $cKsS - C$ to indicate a conv2d layer with $C$ filters of size $[K, K]$ and stride $S$, and $dN$ for a dense layer with $N$ output neurons, the architectures (Arch.) are A: d10; B: c3s2-8, c3s2-16, d10; C: c3s2-32, c3s2-64, d10. The deformations are: [/]: $\boldsymbol{P}_{i,i+1}^j = \boldsymbol{Q}_{i,i+1}^j = \boldsymbol{1}$ (baseline (Peters & Welling, 2018)); [PQ]: $\boldsymbol{P}_{i,i+1}^j, \boldsymbol{Q}_{i,i+1}^j$ generic; [Q]: $\boldsymbol{P}_{i,i+1}^j = \boldsymbol{1}, \boldsymbol{Q}_{i,i+1}^j$ generic.

| Arch. | Deformation | MNIST | Fashion MNIST |
|---|---|---|---|
| A | [/] | 91.1 | 84.2 |
| | [PQ] | **94.3** | **86.8** |
| | [Q] | 91.6 | 85.1 |
| B | [/, /, /] | 96.6 | 87.5 |
| | [PQ, /, /] | **97.6** | **88.1** |
| | [Q, /, /] | 96.8 | 87.8 |
| C | [/, /, /] | 98.1 | 89.3 |
| | [PQ, /, /] | **98.3** | **89.6** |
| | [Q, /, /] | 98.3 | 89.5 |

## 5 CONCLUSIONS

In this work we made the following main contributions: 1) we introduced quantum deformed neural networks and identified potential speedups by running these models on a quantum computer; 2) we devised classically efficient algorithms to train the networks for low entanglement designs of the quantum circuits; 3) for the first time in the literature, we simulated the quantum neural networks on real world data sizes obtaining good accuracy, and showed modest gains due to the quantum deformations. Running these models on a quantum computer will allow one to explore efficiently more general deformations, in particular those that cannot be approximated by the central limit theorem when the Hamiltonians will be sums of non-commuting operators. Another interesting future direction is to incorporate batch normalization and pooling layers in quantum neural networks.

An outstanding question in quantum machine learning is to find quantum advantages for classical machine learning tasks. The class of known problems for which a quantum learner can have a provably exponential advantage over a classical learner is small at the moment Liu et al. (2020), and some problems that are classically hard to compute can be predicted easily with classical machine learning Huang et al. (2020). The approach presented here is the next step in a series of papers that tries to benchmark quantum neural networks empirically, e.g. Farhi & Neven (2018); Huggins et al. (2019); Grant et al. (2019; 2018); Bausch (2020). We are the first to show that towards the limit of entangling circuits the quantum inspired architecture does improve relative to the classical one for real world data sizes.

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
