# OpenReview forum: "Quantum Deformed Neural Networks"
_ICLR.cc/2021/Conference — Reject_

### Official Review · AnonReviewer3 · 2020-10-29
**too many crucial details are deferred to the supplementary material**

**Rating:** 6
**Confidence:** 4

**Review:**

The fundamental idea in this paper is to endow classical signals with a complex Hilberspace structure so as to harness the probabilistic nature of quantum mechaniscs for pattern recognition based on quantum computing principles. Following this idea, the authors consider (probabiistic) binary neural networks and develop a corresponding kind of quantum neural networks building on the concept of quantum phase estimation. They (convincingly) argue that this idea would make good use of the exponential speedup offered by quantum computers but can still be implemented on classical, digital devices. Experiments with image data corroborate this claim and demonstrate that simulations of the proposed quantum deformation showed improved accuracies when compared to  classical probabilistic binary neural networks.

The Ideas brought forth in this paper appear to be novel and (to those with a solid background in quantum computing) technically sound. Overall, the paper presents an interesting approach towards quantum computational intelligence which, though likely not yet realizable on real quantum hardware (due to technical limitations such as measurement noise or decoherence times) can also be simulated digitally (in an arguably elegant manner).

However, there also are concerns regarding the  quality of this manuscript. In additions to several broken LaTeX links (e.g. a reference to Table 4.2 which appears to mean to refer to Table 1) which unnecessarily hamper readability, the overall presentation is lackingcan. Most critically, the paper can hardly be considered self-contained. Numerous important details are deferred to the supplementary material where the authors cover algorithmic details as well as details as to their experimental procedures. In other words, the supplementary material is not just supplementary but crucial for understanding / assessing the content of this work. While this would not justfy an outright rejection, it still diminishes the overall quality of this paper.

---

> ### Author Response · Authors · 2020-11-18
> **State of the art for quantum neural networks**
>
> Thank you for the review. We believe that there is a confusion on what constitutes progress on Quantum Neural Networks (QNNs): our paper advances considerably the SOTA for QNNs, being the first paper to benchmark a quantum model on real world data sizes. While our model might not be competitive against classical ML models, it goes way beyond what has been done in the field of QNNs. To substantiate this claim we provide below a table of papers which benchmark QNNs on MNIST for comparison, showing that our model is the only one that is accurate and can be simulated for real data sizes and 10 classes classification. We ask the reviewer to reconsider their scores in light of this evidence, or to provide examples of QNNs that can be simulated on real world data and are accurate, in which case we would understand the low score.
>
> |ID |Num qubits   |Input dim |Num classes|Accuracy |Note                   |
> |---|-------------|----------|-----------|---------|-----------------------|
> |0  |> 28x28x2    |28x28=784 |10         |98.3     |Inputs + weights qubits|
> |1	|17	          |4x4=16    |2	         |0.7248   |Our implementation     |
> |2	|65           |8x8=64	 |2          |87.6–99.6|                       |
> |3	|10-12	      |2-3 	     |2-10	     |90.8–98.6|PCA,t-SNE preprocessing|
> |4	|10	          |32x32=1024|2	         |82	   |Amplitude encoding,    |
> |   |             |          |           |         |linear discriminator   |
> |5	|8	          |64        |2	         |99.29	   |                       |
> |6	|8	          |8 	     |2    	     |~80% 	   |PCA preprocessing      |
>
> These quantum models are representative of literature of learnable quantum circuits. See also table 1 of paper [3], published in NeurIPS 2020. Refs:
> [0]: Our
> [1]: https://arxiv.org/abs/1802.06002
> [2]: https://arxiv.org/pdf/1803.11537.pdf
> [3]: https://arxiv.org/abs/2006.14619
> [4]: https://arxiv.org/abs/1903.05076
> [5]: https://arxiv.org/pdf/1912.12660.pdf
> [6]: https://www.nature.com/articles/s41534-018-0116-9
>
> Regarding the criticism:
> “several broken LaTeX links “ we have fixed the links.
>
> “the paper can hardly be considered self-contained. Numerous important details are deferred to the supplementary material where the authors cover algorithmic details as well as details as to their experimental procedures”, we will add the algorithm for the classical simulations to the main text.

---

### Official Review · AnonReviewer1 · 2020-11-05
**Interesting model and novel idea, unclear advantage of deformation**

**Rating:** 5
**Confidence:** 4

**Review:**

After rebuttal
Thanks for responding to the concerns in the review.
The main point in the author's rebuttal is that the major contribution of the paper is a novel QNN that can be efficiently simulated. However, the proposed method of simulation is only an approximation of the real run through CLT. The author provided some conceptual justification of this approximation but there is no numerical verification. The approximation can be inaccurate when the variables in the summation are correlated or the number of variables is small. It's unclear how the accuracy of simulation can be guaranteed.

Overall the paper proposes an interesting method of implementing BNN using quantum device, which has potential advantage over classical BNN and may be simulated efficiently with some approximation. More solid justifications of the advantages would improve the quality of the paper.

---------------------------------------------

Originality:
The paper presents a method that uses quantum circuits to realize layers of a stochastic binary neural network, where the activations and weights take binary values, and the weights can be stochastic. It also presents a training method that can be run on a classical computer, based on efficient simulation of specific quantum circuits and approximated computation of the distribution of the activations.
Compared with the classical stochastic binary neural network, the quantum realization may perform efficient "deformation", which creates correlations among the input activations and the weights.
The proposed model is interesting, especially in that it supports stochastic weights and deformation. This model also looks to be new.


Clarity:
The presentation of the model and its quantum realization is clear. The description on the classical simulation is less clear but OK.
The main concerns are
1. There is no justification or discussion on the advantage of deformation.
The advantage of deformation looks to be a way to increase the expressiveness of the model. This is true if the model has only one layer. How much gain can it provide for multi-layer models?
Also, there should be a trade-off between the expressiveness and the realization complexity of the deformation, but this is not discussed.
2. In the classical simulation, there is no discussion on why the chosen form of P and Q can lead to better performance. Also, the details of P and Q are not revealed.
3. It is not clear why a modification as in Section 3.3 is needed for classical simulation.
4. In Section 2, the formulation of the training objective (2) is also unclear. It claims that the weights can be represented by a Bayesian posterior and trained by ELBO, but (2) is different from ELBO.


Significance:
In the experiment session, it is reported that the performance on MNIST is better with deformation than without it.
But this comparison may be trivial, as deformation adds more parameters into the plain model.
Also, this may not mean that the proposed model when used as a classical model can perform better than the classical binary neural network, as the proposed model can have much higher computational complexity for performing the deformation.
It would be more convincing if some comparison can be made between the proposed model and the other quantum models or classical models with similar complexity.

---

> ### Author Response · Authors · 2020-11-18
> **State of the art for quantum neural networks**
>
> Thank you for the review. We believe that there is a confusion on what constitutes progress on Quantum Neural Networks (QNNs): our paper advances considerably the SOTA for QNNs, being the first paper to benchmark a quantum model on real world data sizes. While our model might not be competitive against classical ML models, it goes way beyond what has been done in the field of QNNs. To substantiate this claim we provide below a table of papers which benchmark QNNs on MNIST for comparison, showing that our model is the only one that is accurate and can be simulated for real data sizes and 10 classes classification. We ask the reviewer to reconsider their scores in light of this evidence, or to provide examples of QNNs that can be simulated on real world data and are accurate, in which case we would understand the rejection.
>
> |ID |Num qubits   |Input dim |Num classes|Accuracy |Note                   |
> |---|-------------|----------|-----------|---------|-----------------------|
> |0  |> 28x28x2    |28x28=784 |10         |98.3     |Inputs + weights qubits|
> |1	|17	          |4x4=16    |2	         |0.7248   |Our implementation     |
> |2	|65           |8x8=64	 |2          |87.6–99.6|                       |
> |3	|10-12	      |2-3 	     |2-10	     |90.8–98.6|PCA,t-SNE preprocessing|
> |4	|10	          |32x32=1024|2	         |82	   |Amplitude encoding,    |
> |   |             |          |           |         |linear discriminator   |
> |5	|8	          |64        |2	         |99.29	   |                       |
> |6	|8	          |8 	     |2    	     |~80% 	   |PCA preprocessing      |
>
> These quantum models are representative of literature of learnable quantum circuits. See also table 1 of paper [3], published in NeurIPS 2020. Refs:
> [0]: Our
> [1]: https://arxiv.org/abs/1802.06002
> [2]: https://arxiv.org/pdf/1803.11537.pdf
> [3]: https://arxiv.org/abs/2006.14619
> [4]: https://arxiv.org/abs/1903.05076
> [5]: https://arxiv.org/pdf/1912.12660.pdf
> [6]: https://www.nature.com/articles/s41534-018-0116-9
>
> We now address the concerns raised:
>
> 1) “There is no justification or discussion on the advantage of deformation.” We have run experiments comparing the quantum deformed models against classical nets with the same number of parameters by increasing the number of intermediate channels in the conv layers. These results show that the quantum models still deliver slightly better accuracies than classical models with the same number of parameters, see table below (Arch. n,m means two conv layers with n,m channels resp. Cf arch B of the table 1 in the paper):
>
> |Arch. |Deformation |Num params|MNIST  | Fashion MNIST |
> |------|------------|----------|-------|---------------|
> |8, 16 |[/, /, /]   |7018      |96.6   |87.5           |
> |8, 22 |[/, /, /]   |9616      |96.7   |88.1           |
> |9, 21 |[/, /, /]   |9382      |96.7   |87.9           |
> |10, 21|[/, /, /]   |9581      |97.1   |87.9           |
> |12, 20|[/, /, /]   |9510      |97.1   |88.1           |
> |8, 16 |[PQ, /, /]  |9322      |97.6   |88.1           |
>
> We will update the paper with these results.
>
> 2) “there is no discussion on why the chosen form of P and Q can lead to better performance. Also, the details of P and Q are not revealed”, we choose P,Q as arbitrary 4x4 unitary matrices. We parameterize the logarithm of P,Q as anti-Hermitian matrices (16 real parameters each) and use the matrix exponential to compute P,Q. We will add the details to the paper. Also, we will add the algorithm for the classical simulations to the main text.
>
> 3) “It is not clear why a modification as in Section 3.3 is needed for classical simulation.“ In the proposal of figure 1 (a/b)  we perform the computations for different hidden neurons sequentially. In the modification of Fig 1 (c) we clone the input state and process the hidden neurons in parallel, which gives higher efficiency for the classical simulations.
>
> 4) “training objective (2) is also unclear” , we agree that formula (2) is wrong. We will fix the formula and the discussion using the right bound which has E(log p) (as appears in the ELBO) instead of log E(p). Note that however this does not change our training procedure, simply fixes a mistake on the bound.
>
> “It would be more convincing if some comparison can be made between the proposed model and the other quantum models or classical models with similar complexity.” Please refer to the two table above for comparisons with other quantum models and classical models with similar complexity. These show that the model we are proposing is competitive with classical models and greatly outperforms the other quantum models in the literature (input size differs by two orders of magnitude).

---

### Official Review · AnonReviewer7 · 2020-11-06
**Interesting idea, but practical and theoretical benefits seem marginal**

**Rating:** 4
**Confidence:** 4

**Review:**

__Summary__
A "quantum deformed" generalization of a probabilistic binary neural network is introduced, which can be either run on a quantum computer or (for certain simplified cases) simulated with a classical computer. The performance of the model under classical simulation is assessed with experiments on the MNIST and Fashion MNIST datasets.

__Strengths__
The fusion of primitives from quantum computing with a variational Bayesian framework is interesting, and seems like a promising general means of leveraging the complementary strengths of ML and quantum computing.

The proposed model can be implemented in two different "modes of operation", either as a fully quantum model running on a quantum computer or as a classical simulation procedure running on a classical (i.e. standard modern-day) computer. Hybrid methods with this capability for dual use seem promising for quantum ML, since they allow the same framework to be gracefully transitioned from classical computers to quantum computers as the capabilities of the latter improve with time.

__Weaknesses__
The practical benefits of the proposed model aren't clearly explained, and the merits of the architecture appear rather weak to me, both from the standpoints of ML and from that of quantum computing. I will elaborate on this in the following paragraphs, but a short summary is that the work (a) Doesn't give any evidence that these methods help in learning real-world ML datasets relative to standard neural networks, and (b) Doesn't list any advantages of the proposed model for the implementation of quantum computing protocols. I'm open to the possibility of error in this assessment, and would love if the authors could provide clear evidence refuting either of these two points.

From the perspective of (classical) ML, I can't see any notable advantages of the proposed methods. The authors report small gains relative to the previous proposal of [1] (although see the last point below), but the primary motivation of [1] was to design a lightweight binarized neural network with reduced compute and memory costs. The present work doesn't do this, with the classical mode of operation requiring significant resources, and also delivering performance below a CNN of the same size.

The authors frequently reference the "quantum supremacy" work of [2] and [3] as evidence of the advantages of quantum computers, and mention that the proposed model could be used for such tasks. However, quantum supremacy experiments are not designed to be useful for real-world problems (see [4] for a detailed discussion regarding this point). Rather, the primary role of quantum supremacy experiments is to unambiguously certify that a resource limited quantum computer/simulator is doing *something* that can't be efficiently reproduced with a classical computer, even if that something has no practical use. As a result, any proposed methods for quantum supremacy should be judged by their theoretical guarantees of classical insimulatability, along with their ease of implementation in noisy present-day quantum computing hardware. With that in mind...

The present work doesn't give any new theoretical guarantees that the proposed architecture demonstrates quantum supremacy, instead briefly mentioning that the quantum circuits proposed in [2] and [3] could be embedded inside the current model (making the current proposal strictly more difficult to implement than these prior works). The use of quantum phase estimation throughout the quantum deformed neural network means that the proposed algorithm would be very challenging to implement in practice, likely only being possible when quantum hardware has advanced to the point that such supremacy demonstrations are no longer a subject of interest.

The MNIST and Fashion MNIST experiments are rather weak, with the only baseline being the closely related model of [1]. Relative to standard neural networks, these results are far from state-of-the-art, and the classically simulable version of the protocol is also significantly more expensive to run than a basic MLP or CNN.

One point of confusion I ran into is the experimental results reported for the baseline model of [1]. The values reported in the present paper for this baseline are significantly lower than the MNIST results originally reported in [1], with the latter giving better results than anything exhibited by the present architecture. Can the authors comment on this discrepancy?

Overall assessment:
Although the present work provides some interesting ideas, the theoretical and practical benefits of the model seem marginal. Consequently, I can't recommend acceptance.

[1] Jorn W.T. Peters and Max Welling. Probabilistic binary neural networks, 2018.
[2] Scott Aaronson and Lijie Chen. Complexity-theoretic foundations of quantum supremacy experiments, 2016.
[3] Frank Arute et al. Quantum supremacy using a programmable superconducting processor. Nature 574, 2019.
[4] Scott Aaronson. Scott’s Supreme Quantum Supremacy FAQ, 2019. (https://www.scottaaronson.com/blog/?p=4317)

### UPDATE AFTER THE REBUTTAL

Many thanks to the authors for their reply, and my sincere apologies for forgetting to include my bibliography in the earlier response (you can find it above)! I also appreciate the information about the comparison to [1], as well as the promise of a clarifying statement about the appropriate experimental context for the use of quantum phase estimation.

While I agree that the comparison to previous QNN models is favorable, I still think that the use of quantum phase estimation in the proposed method is problematic, as it would significantly delay any potential deployment of the proposed method on a quantum computer. I don't disagree with the authors that their statement regarding the ability to demonstrate quantum supremacy is technically correct, but they haven't addressed my question about _why_ there would be any advantage to carrying out a quantum supremacy experiment on such a model. Studying QNNs for the sake of QNNs isn't a compelling justification in my eyes, and without any strong practical advantages (either definite or potential) provided for the model, my rating unfortunately remains the same.

---

> ### Author Response · Authors · 2020-11-18
> **State of the art for quantum neural networks**
>
> Thank you for the review. We believe that there is a confusion on what constitutes progress on Quantum Neural Networks (QNNs): our paper advances considerably the SOTA for QNNs, being the first paper to benchmark a quantum model on real world data sizes. While our model might not be competitive against classical ML models, it goes way beyond what has been done in the field of QNNs. To substantiate this claim we provide below a table of papers which benchmark QNNs on MNIST for comparison, showing that our model is the only one that is accurate and can be simulated for real data sizes and 10 classes classification. We ask the reviewer to reconsider their scores in light of this evidence, or to provide examples of QNNs that can be simulated on real world data and are accurate, in which case we would understand the rejection.
>
> |ID |Num qubits   |Input dim |Num classes|Accuracy |Note                   |
> |---|-------------|----------|-----------|---------|-----------------------|
> |0  |> 28x28x2    |28x28=784 |10         |98.3     |Inputs + weights qubits|
> |1	|17	          |4x4=16    |2	         |0.7248   |Our implementation     |
> |2	|65           |8x8=64	 |2          |87.6–99.6|                       |
> |3	|10-12	      |2-3 	     |2-10	     |90.8–98.6|PCA,t-SNE preprocessing|
> |4	|10	          |32x32=1024|2	         |82	   |Amplitude encoding,    |
> |   |             |          |           |         |linear discriminator   |
> |5	|8	          |64        |2	         |99.29	   |                       |
> |6	|8	          |8 	     |2    	     |~80% 	   |PCA preprocessing      |
>
> These quantum models are representative of literature of learnable quantum circuits. See also table 1 of paper [3], published in NeurIPS 2020. Refs:
> [0]: Our
> [1]: https://arxiv.org/abs/1802.06002
> [2]: https://arxiv.org/pdf/1803.11537.pdf
> [3]: https://arxiv.org/abs/2006.14619
> [4]: https://arxiv.org/abs/1903.05076
> [5]: https://arxiv.org/pdf/1912.12660.pdf
> [6]: https://www.nature.com/articles/s41534-018-0116-9
>
> We now address the weaknesses raised:
>
> “From the perspective of (classical) ML, I can't see any notable advantages of the proposed methods”, “Relative to standard neural networks, these results are far from state-of-the-art”: As remarked above, we achieve SOTA relative to QNN models, which is the focus of our work. Regarding “The values reported in the present paper for this baseline are significantly lower than the MNIST results originally reported in [1]”, assuming that [1] is [Peters, Welling 2018], the difference is due to the different models: we use fewer layers to make the simulation of the deformation cheaper and use no batch norm, and no max pooling. We will add a comment in the manuscript. To provide a fairer comparison with classical models, we have run experiments comparing the quantum deformed models against classical nets with the same number of parameters by increasing the number of intermediate channels in the conv layers. These results show that the quantum models still deliver slightly better accuracies than classical models with the same number of parameters, see table below (Arch. n,m means two conv layers with n,m channels resp. Cf arch B of the table 1 in the paper):
>
> |Arch. |Deformation |Num params|MNIST  | Fashion MNIST |
> |------|------------|----------|-------|---------------|
> |8, 16 |[/, /, /]   |7018      |96.6   |87.5           |
> |8, 22 |[/, /, /]   |9616      |96.7   |88.1           |
> |9, 21 |[/, /, /]   |9382      |96.7   |87.9           |
> |10, 21|[/, /, /]   |9581      |97.1   |87.9           |
> |12, 20|[/, /, /]   |9510      |97.1   |88.1           |
> |8, 16 |[PQ, /, /]  |9322      |97.6   |88.1           |
>
> We will update the paper with these results.
>
> “The present work doesn't give any new theoretical guarantees that the proposed architecture demonstrates quantum supremacy”, Our supremacy statement is not wrong because we claim computational aspects for certain choices of the circuit D of the paper (e.g. random 2d circuits, physical Hamiltonians). We do not claim better learning performance and do not exclude that there can be another classical architecture which gets better accuracy for a given ML task. Our approach is to get as close as possible to a quantum model with something that you can simulate classically. We are the first to show that towards the limit of entangling D the quantum inspired architecture does improve relative to the classical one. Our work is the next step in a tradition of papers that tries to benchmark QNNs empirically. With respect to implementation on quantum hardware, we will add a comment to clarify that we are not proposing to implement quantum phase estimation on NISQ devices and the general architecture will require error corrected quantum computers.

---

### Official Review · AnonReviewer6 · 2020-11-08
**An clearly-written and interesting attempt to define a useful neural network that can run on a quantum computer, but not enough evidence is provided for its efficacy.**

**Rating:** 4
**Confidence:** 4

**Review:**

The authors define a quantum version of a binary neural network (all weights and activations are 0 or 1) by first defining a classical stochastic generalization and then upgrading the stochastic part to a quantum process. The calculations all appear to be correct, and to my knowledge this type of quantum neural network is novel. The quantum network is simulated on MNIST and Fashion MNIST. The quantum network shows improvements in test accuracy, but those improvements are attributed to increased parameterization. Thus the jury is still out on whether this is actually an interesting use-case for quantum computers, and it's not clear that this work will have any real impact on the field. For that reason, I cannot recommend this paper for acceptance.

Pros:
1) Clearly explains the quantum theory in a way that should be understandable to a novice.
2) A novel design of quantum neural networks, to the best of my knowledge.
3) Makes an attempt to test the performance in simulation.

Cons:
1) The authors make a number of claims about possible quantum advantage, but there is no reason to suspect quantum advantage based on the evidence presented. (Generally, one should expect that there is no quantum advantage for classical data processing unless clear evidence is given to the contrary.) It would be good to attempt to distinguish between improvements in performance due to overparameterization from improvements due to quantum properties, e.g., entanglement.
2) By the same token, is there any benefit from using the quantum deformed model over the classical stochastic model? More information about the stochastic model would be helpful.
3) The paper would benefit from a discussion of near-term vs long-term uses of a quantum computer (i.e., noisy vs error-corrected). The authors use phase estimation in their construction, which is typically an operation that is reserved for error-corrected quantum computers. The authors mention noisy intermediate-scale computers at one point, but it's not clear at all that this algorithm is applicable in that case. If it is, the authors should discuss the effects of noise on their results.


Miscellaneous comments:
1) The authors cite Hooft 2016, but this should be 't Hooft. The 't part is included with the surname.
2) In the 4th paragraph of the introduction, "restrict" should be "restricted".
3) The authors state that Farhi and Neven 2018 restricted themselves to 4x4 images because of exponential time of simulations. I believe it's more likely that it was memory constraints of doing full quantum simulations that was important there.
4) in the second paragraph of section 2 there is a malformed citation that just says "staines2012variational".
5) In section 3.1, the authors state that a qubit is a normalized vector. A qubit more properly refers to the vector space, or to the set of unit vectors in that space, rather than an individual vector. The wording is a bit sloppy, since N qubits are properly associated with a vector space in the following line.
6) In the last paragraph of 3.2, the authors refer to the V_h subspace. V_h is not a subspace, but a factor (or subfactor). The distinction is important as it often confuses people.
7) In the first paragraph of 3.3, "it's' should be "its".

---

> ### Author Response · Authors · 2020-11-18
> **State of the art for quantum neural networks**
>
> Thank you for the review. We believe that there is a confusion on what constitutes progress on Quantum Neural Networks (QNNs): our paper advances considerably the SOTA for QNNs, being the first paper to benchmark a quantum model on real world data sizes. While our model might not be competitive against classical ML models, it goes way beyond what has been done in the field of QNNs. To substantiate this claim we provide below a table of papers which benchmark QNNs on MNIST for comparison, showing that our model is the only one that is accurate and can be simulated for real data sizes and 10 classes classification. We ask the reviewer to reconsider their scores in light of this evidence, or to provide examples of QNNs that can be simulated on real world data and are accurate, in which case we would understand the rejection.
>
> |ID |Num qubits   |Input dim |Num classes|Accuracy |Note                   |
> |---|-------------|----------|-----------|---------|-----------------------|
> |0  |> 28x28x2    |28x28=784 |10         |98.3     |Inputs + weights qubits|
> |1	|17	          |4x4=16    |2	         |0.7248   |Our implementation     |
> |2	|65           |8x8=64	 |2          |87.6–99.6|                       |
> |3	|10-12	      |2-3 	     |2-10	     |90.8–98.6|PCA,t-SNE preprocessing|
> |4	|10	          |32x32=1024|2	         |82	   |Amplitude encoding,    |
> |   |             |          |           |         |linear discriminator   |
> |5	|8	          |64        |2	         |99.29	   |                       |
> |6	|8	          |8 	     |2    	     |~80% 	   |PCA preprocessing      |
>
> These quantum models are representative of literature of learnable quantum circuits. See also table 1 of paper [3], published in NeurIPS 2020. Refs:
> [0]: Our
> [1]: https://arxiv.org/abs/1802.06002
> [2]: https://arxiv.org/pdf/1803.11537.pdf
> [3]: https://arxiv.org/abs/2006.14619
> [4]: https://arxiv.org/abs/1903.05076
> [5]: https://arxiv.org/pdf/1912.12660.pdf
> [6]: https://www.nature.com/articles/s41534-018-0116-9
>
> We now address the cons raised:
>
> 1) , 2. “distinguish between improvements in performance due to overparameterization from improvements due to quantum properties” and “More information about the stochastic model”: we have run experiments comparing the quantum deformed models against classical nets with the same number of parameters by increasing the number of intermediate channels in the conv layers. These results show that the quantum models still deliver slightly better accuracies than classical models with the same number of parameters, see table below (Arch. n,m means two conv layers with n,m channels resp. Cf arch B of the table 1 in the paper):
>
> |Arch. |Deformation |Num params|MNIST  | Fashion MNIST |
> |------|------------|----------|-------|---------------|
> |8, 16 |[/, /, /]   |7018      |96.6   |87.5           |
> |8, 22 |[/, /, /]   |9616      |96.7   |88.1           |
> |9, 21 |[/, /, /]   |9382      |96.7   |87.9           |
> |10, 21|[/, /, /]   |9581      |97.1   |87.9           |
> |12, 20|[/, /, /]   |9510      |97.1   |88.1           |
> |8, 16 |[PQ, /, /]  |9322      |97.6   |88.1           |
>
> We will update the paper with these results.
> Further, we remark that our supremacy statement is about computational aspects. We do not claim better learning performance and do not exclude that there can be another classical architecture which gets better accuracy for a given ML task. Our approach is to get as close as possible to a quantum model with something that you can simulate classically. We are the first to show that towards the limit of entangling D the quantum inspired architecture does improve relative to the classical one. Our work is the next step in a sequence of papers that tries to benchmark QNNs empirically.
>
> 3) “authors mention noisy intermediate-scale computers at one point, but it's not clear at all that this algorithm is applicable in that case”, we mention it as evidence that one can use NISQ devices to sample more efficiently than classically for certain distributions. We will add a comment to clarify that we are not proposing to implement quantum phase estimation on NISQ devices.
>
> We thank you for the miscellaneous comments and will fix them.

---

### Official Review · AnonReviewer4 · 2020-11-13
**An interesting work on building novel quantum neural networks and the corresponding quantum-inspired neural networks; however without establishing any clear quantum or classical advantages**

**Rating:** 6
**Confidence:** 4

**Review:**

The authors introduced the idea of quantum deformed neural network (DQNN) which allows substitution of the positive probabilities in Baysian statistics with complex amplitude inspired by Born representation of quantum wavefunction, thus allowing interference phenomena leading to possible speed up for running neural networks on quantum computers. Combining the complex amplitude and binary neural network the author introduces a new class of quantum neural networks which are the generalized probabilistic neural network. Then, they argue that this class of DQNN can efficiently run on quantum computers using phase estimation algorithms which scale linearly with the number of qubits. By creating entanglement between the activation and weights they show that this new form of DQNN can be simulated on classical computers for certain classes of problems that hold low entanglement. As a numerical experiment they show modest gain in accuracy on real world data both MNIST and fashion MNITS data.

Pro:
Novel encoding: in this work the two main properties of quantum mechanics namely the interference and entanglement is being encoded in the deformed neural network to investigate potential quantum speed up within this platform

Cons:
1- The author loosely discuss the expressive power of the DQNN compared to NN.

2. The implementation of phase estimation algorithm requires large depth (linear in number of qubits) and considering the fact that in this algorithm the number of required qubits increases by increasing the connectivity of the neural network, the quantum implementation becomes quite challenging and beyond NISQ device. It probably would be disadvantages compare to the QNN in Farhi's paper.

3. The author also compare their work with tensor network, however, they seem not to be updated on what has been done on this part and they make a few inaccurate claims:
     3A. On page 2, section 1.1 on second paragraph the author provides a brief study on the tensor network, claiming that "However the constraints on the network geometry to allow for efficient contractions limit the expressivity and performance of the models. Further these works do not directly relate to implementations on quantum computers".
this is not a correct statement as there are direct implementation of tensor network on quantum circuit, see ref [Huggins], also in recent paper [https://arxiv.org/abs/1907.03741] led by Cirac, they provide proof that quantum circuits are Born machines or locally purified states.
     3B. The author also claims that their study is one of the first studied in QML where they are able to work on the real size MNIST data, which again is not true. Indeed, Lei Wang and his collaborators in Ref[https://arxiv.org/abs/2009.09932] were able to use supervised learning on PEPS for MNIST and MNIST fashion data.

4. Also, while the tensor network enjoys a concrete parametrization of entanglement with the bond dimension, it's less clear for me how they can tune the entanglement within their formalism.

5. The authors have multiple referral to exponential advantage (or quantum supremacy) in quantum computers which are either incorrect or at least misleading in this context. Those claims are relevant for either random quantum circuits (which have no clear applications) or quantum simulations of certain physically sparse Hamiltonians. In general, the learning complexity could be  independent of simulation complexity of quantum computers once the data is provided, for example see the recent work on the Power of data in quantum machine learning: arXiv:2011.01938

6. The authors conclusion about limited advantage of quantum-inspired deformed neural networks is not correct as it could be simply explained by the over parametrization of deformed model

---

> ### Author Response · Authors · 2020-11-18
> **State of the art for quantum neural networks**
>
> Thank you for the review. We believe that there is a confusion on what constitutes progress on Quantum Neural Networks (QNNs): our paper advances considerably the SOTA for QNNs, being the first paper to benchmark a quantum model on real world data sizes. While our model might not be competitive against classical ML models, it goes way beyond what has been done in the field of QNNs. To substantiate this claim we provide below a table of papers which benchmark QNNs on MNIST for comparison, showing that our model is the only one that is accurate and can be simulated for real data sizes and 10 classes classification. We ask the reviewer to reconsider their scores in light of this evidence, or to provide examples of QNNs that can be simulated on real world data and are accurate, in which case we would understand the rejection.
>
>
> |ID |Num qubits   |Input dim |Num classes|Accuracy |Note                   |
> |---|-------------|----------|-----------|---------|-----------------------|
> |0  |> 28x28x2    |28x28=784 |10         |98.3     |Inputs + weights qubits|
> |1	|17	          |4x4=16    |2	         |0.7248   |Our implementation     |
> |2	|65           |8x8=64	 |2          |87.6–99.6||
> |3	|10-12	      |2-3 	     |2-10	     |90.8–98.6|PCA,t-SNE preprocessing|
> |4	|10	          |32x32=1024|2	         |82	   |Amplitude encoding,    |
> |   |             |          |           |         |linear discriminator   |
> |5	|8	          |64        |2	         |99.29	   |       |
> |6	|8	          |8 	     |2    	     |~80% 	   |PCA preprocessing      |
>
> These quantum models are representative of literature of learnable quantum circuits. See also table 1 of paper [3], published in NeurIPS 2020. Refs:
> [0]: Our
> [1]: https://arxiv.org/abs/1802.06002
> [2]: https://arxiv.org/pdf/1803.11537.pdf
> [3]: https://arxiv.org/abs/2006.14619
> [4]: https://arxiv.org/abs/1903.05076
> [5]: https://arxiv.org/pdf/1912.12660.pdf
> [6]: https://www.nature.com/articles/s41534-018-0116-9
>
> We now address the cons raised:
> 1) , 6. We have run experiments comparing the quantum deformed models against classical nets with the same number of parameters by increasing the number of intermediate channels in the conv layers. These results show that the quantum models still deliver slightly better accuracies than classical models with the same number of parameters (Arch. n,m means two conv layers with n,m channels resp. Cf arch B of the table 1 in the paper):
>
> |Arch. |Deformation |Num params|MNIST  | Fashion MNIST |
> |------|------------|----------|-------|---------------|
> |8, 16 |[/, /, /]   |7018     |96.6   |87.5 |
> |8, 22 |[/, /, /]   |9616     |96.7   |88.1 |
> |9, 21 |[/, /, /]   |9382     |96.7   |87.9 |
> |10, 21|[/, /, /]  |9581     |97.1   |87.9 |
> |12, 20|[/, /, /]  |9510     |97.1   |88.1 |
> |8, 16 |[PQ, /, /]|9322    |97.6   |88.1  |
>
> We will update the paper with these results.
>
> 2) We will add a comment to clarify that we are not proposing to implement quantum phase estimation on NISQ devices. We cannot compare directly to Farhi-Neven since they only discuss their architecture for 4x4 images, while we address full size images. Nonetheless, we implemented the Farhi-Neven model as faithfully as possible (we also emailed the authors to ask for their code, but they could not share it) on 4x4 images, getting training/test accuracy 0.9935/0.7248 with 96 parameters.  Instead we got 0.9997/0.9978 for the simplest QNN of our paper with 91 parameters.
>
> 3) We respectfully disagree that our claim is incorrect. We comment on the fact that given a generic tensor network it is not clear in general how to associate to it a quantum circuit. Instead [Huggins] is about a specific tensor network with unitary tensors, [https://arxiv.org/abs/1907.03741] is about interpreting quantum circuits as tensor networks. All the models we refer to (Miles Stoudenmire & Schwab, 2016; Liu et al., 2017; Levine et al., 2017; 2019), as well as the suggested [https://arxiv.org/abs/2009.09932], are tensor networks but not quantum circuits, and our model is therefore the first quantum neural network to run on real world data, as claimed.
>
> 4) The circuit D controls how inputs and weights are entangled. In our experiments we take D to be a 1d quantum circuit, and the time evolution under D can be carried out using MPS techniques (TEBD), allowing one to relate directly the depth of D to the MPS bond dimension.
>
> 5) Our supremacy statement is not wrong because we only make claims about computational aspects. Further, we do not see why a priori we need to exclude random 2d circuits or physical Hamiltonians as possible D’s. We do not claim better learning performance and do not exclude that there can be another classical architecture which gets better accuracy for a given ML task.  Our work is the next step in a tradition of papers that tries to benchmark QNNs empirically. We read arXiv:2011.01938 but did not find evidence that our architecture is in the class of models for which classical ML models behave better.

---

### Decision · Program_Chairs · 2021-01-07
**Final Decision**

**Decision:**

Reject

**Comment:**

 A "quantum deformed" generalization of a probabilistic binary neural network is introduced, which can be either run on a quantum computer or simulated with a classical computer. Reviewers agreed that the paper is well written, introduces some new ideas merging quantum computing with a variational Bayesian framework, and the reported numbers on MNIST and Fashion MNIST outperform prior QNN approachers. However, reviewers questioned how useful the proposed ideas are, noting that: the reported gains could be attributed to increased parameterization (this was not carefully ablated with baseline approaches). Additionally, while the quantum supremacy experiments seem technically correct, there was no clear motivation for empirically demonstrating quantum supremacy when no theoretical guarantees are provided. Taken together, there was no clear path to practical improvements of real systems from the proposed ideas.